# Effect of Deep Cryogenic Treatment on the Artificial Ageing Behavior of SiCp–AA2009 Composite

Zhenxiao Wang [1], Jie Chen [1], Baosheng Liu [2], Ran Pan [2,*], Yuan Gao [1] and Yong Li [1,3,*]

1   School of Mechanical Engineering and Automation, Beihang University, Beijing 100191, China
2   AVIC Manufacturing Technology Institute, Beijing 100024, China
3   Shenzhen Institute of Beihang University, Shenzhen 518057, China
*   Correspondence: bkdpanran@163.com (R.P.); liyong19@buaa.edu.cn (Y.L.)

**Abstract:** The effect of deep cryogenic treatment (DCT) on the artificial ageing kinetics of a SiC particles reinforced aluminum alloy composite (SiCp-Al) is experimentally studied in this paper. The evolutions of both macro-properties (i.e., yield strength and ultimate tensile strength) and microstructures (precipitates) have been investigated by a set of hardness tests, tensile tests, and microstructural observations (scanning electron microscope, SEM and transmission electron microscope, TEM) for a SiCp-Al composite material with conventional heat treatment (solution heat treatment + quenching + artificial ageing, CHT) or DCT (solution heat treatment + quenching + deep cryogenic + artificial ageing). The results show that SiCp could significantly accelerate the ageing kinetics of the composites. Meanwhile, compared with CHT conditions, DCT can further improve the yield strength (YS) and ultimate tensile strength (UTS) of the composite materials after artificial ageing. The microstructures show that DCT induces the generation of more thinner θ′ precipitates homogeneously distributed in the grains during artificial ageing compared with corresponding CHT conditions. A quantified analysis has been performed based on the microstructural data, and the calculated results further support the indication that the strengthening effect in DCT compared with CHT is mainly contributed by corresponding precipitation behavior.

**Keywords:** deep cryogenic treatment; SiCp reinforced aluminum alloy composite; artificial ageing; precipitation; strength

## 1. Introduction

Metal–matrix composites (MMCs) are becoming a popular choice for the lightweight products in the aerospace industry due to their excellent specific strength and stiffness properties. Silicon carbide particles (SiCp) reinforced aluminum alloy composite (SiCp-Al), which possesses good low thermal expansion properties, is one of the most promising MMCs that have attracted high interest in both academia and industry. It has been widely reported that reinforced SiCp could increase the corrosion resistance [1], yield strength (YS) [2], ultimate tensile strength (UTS) [3] and hardness [4] of composites. Furthermore, SiCp may affect the flow behavior in tensile tests by the occurrence of Portevin-le Chatelier (PLC) effects [5–7]. Similar to the matrix material of aluminum alloys, SiCp-Al composites have a strengthening capability by ageing treatment. However, the ageing dynamics of composites would be changed when compared with that of matrix Al material due to the influence of the micron particle reinforced phase. Kim et al. [8], Starink and Gregson [9] as well as Härtel et al. [10] have reported the effect of SiCp on the acceleration effect of age hardening. Meanwhile, it is still a popular research topic to investigate the novel method to further improve the strength of the composite materials. A potential method by adding deep cryogenic treatment to the processes route of materials has been tested for various ferrous alloys [11], and recently also for aluminum alloys [12], to explore its potential of better mechanical properties of the materials. Hence, it is of great interest in



exploring the potential strengthening effect and mechanisms of cryogenic treatment on the SiCp-Al composites.

Deep cryogenic treatment (DCT) has been successfully applied in steels [13–17] and non-ferrous metals such as titanium alloys [18–20] and aluminum alloys [21–23]. Zhou et al. [22] have found that DCT could reduce the residual stress of aluminum alloy components by 72.7%, and proposed a possible reason that the residual stress generated during the sample's re-heating from cryogenic temperature may partly balance the existing residual stress after quenching. Pankaj et al. [24] have reported that DCT could increase the strength of the aluminum alloys after ageing by 29%. Araghchi et al. [25] have studied the properties of S′ phase ($Al_2CuMg$) of 2024 aluminum alloy samples after DCT, and found that DCT increased dislocation density, which provided more nucleation sites for S′ phase. This method may provide a potential way to further enhance the strength of the metal matrix composites; however, the detailed effects and mechanisms of DCT on ageing kinetics of metal matrix composites are not investigated and reported currently, which is of great interest for both academia and industry.

Hence, the effect of DCT on subsequent ageing kinetics of a SiCp-Al composite, 15 vol.% SiCp–AA2009 composite, has been investigated in this study. Key mechanical properties, including yield strength (YS), ultimate tensile strength (UTS) and microstructure analysis, were performed to investigate the macro behavior and corresponding mechanisms for the effect of DCT on ageing hardening. Meanwhile, theoretical calculations have also been carried out to relate the evolution of microstructures and corresponding strength, providing quantified support for the analysis.

## 2. Materials and Methods

### 2.1. Materials and Heat Treatment

An Al-Cu-Mg alloy (AA2009) with compositions listed in Table 1 was selected as the base material for the SiCp reinforced composites investigated in this study, in which 15 vol.% SiCp were uniformed mixed. The materials were provided in an as-forged state by the Institute of Metal Research, Chinese Academy of Sciences (IMR), whose detailed manufacturing processes were introduced before [26].

**Table 1.** Chemical compositions of AA2009 aluminum alloy (wt%).

| Cu | Mg | Si | Fe | Zn | O | Impurity | Al |
|------|------|------|------|------|------|----------|---------|
| 3.44 | 1.38 | 0.29 | 0.06 | 0.05 | 0.15 | 0.15 | Balance |

Conventional solution heat treatment (SHT) and artificial ageing tests were performed for both the AA2009 base material and corresponding SiCp–AA2009 composites to compare their ageing behaviors. In addition, a deep cryogenic treatment (DCT) was inserted between the SHT and artificial ageing treatments for the composite materials to investigate the DCT effect. Figure 1 illustrates the detailed heat treatment process. SHT was carried out at 510 °C ($T_S$, solution temperature) for 2 h with subsequent water-quenching. For conventional treatment, quenched samples were artificially aged at 170 °C ($T_A$, age temperature) in a heat treatment furnace with different ageing time. However, for the DCT treatment, the quenched samples were immersed in the liquid nitrogen directly, further cooling to −196 °C ($T_{DCT}$, deep cryogenic treatment temperature). After DCT for a designed time, samples were taken out and placed into the furnace to perform the artificial ageing tests.

The detailed experimental plan is listed in Table 2. The first and second group of tests for composites and base materials respectively were performed with conventional heat treatment strategy (CHT), and the hardness measurement was performed after different ageing times to obtain the hardness evolutions of both materials during CHT. Based on the results of hardness tests, the 3rd group of tests with CHT strategy was designed to obtain the basic mechanical properties of the composite material with different artificial ageing

times by tensile tests, while corresponding tests with DCT strategy were carried out with an intermediate 1 h DCT treatment between SHT and artificial ageing, as listed in Table 2.

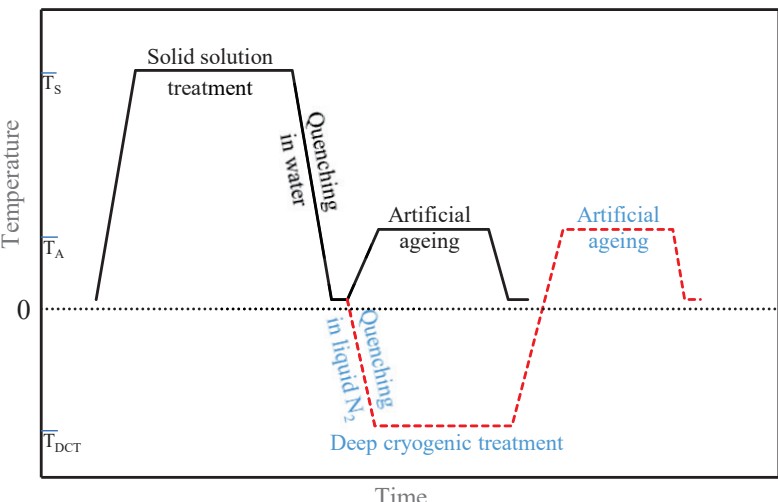

**Figure 1.** Schematic showing the heat treatment route.

**Table 2.** Experimental plan.

| Group | Material | SHT | DCT | Artificial Ageing (170 °C, $T_A$) | Tests |
|---|---|---|---|---|---|
| 1 | Composites | 510 °C ($T_S$) for 2 h | - | 0–60 h | Hardness |
| 2 | AA2009 | 510 °C ($T_S$) for 2 h | - | 0–60 h | Hardness |
| 3 | Composites | 510 °C ($T_S$) for 2 h | - | 0–60 h | Tensile |
| 4 | Composites | 510 °C ($T_S$) for 2 h | −196 °C ($T_{DCT}$) for 1 h | 0–60 h | Tensile |

### 2.2. Mechanical and Microstructural Tests

The as-forged materials were cut into cubic samples with a length of 10 mm for hardness tests. All samples were ground perpendicular to the forging direction prior to measurement. Hardness tests were performed using an INNOVATEST's Flacon-500 Vickers hardness tester. Samples were applied a load pressure of 200 gf and held for 10 s following GB/T 4340-2009 standards. At least ten points were randomly selected in each sample for hardness measurement to obtain the average hardness values.

Dog-bone samples were prepared for tensile tests, as shown in Figure 2. Tensile tests were performed using an electronic universal testing machine at room temperature. An extensometer with a gauge length of 25 mm was used to capture the strain data during tensile tests, with which the yield strength with 0.2% offset, tensile strength and elongation at failure can be calculated. The strain rate used in the tests was $1 \times 10^{-3}$ s$^{-1}$ following GB/T 228.1-2021 standards. Repeated tests were carried out for some selected samples to confirm the repeatability of the results.

In order to investigate the detailed mechanisms of the strength evolutions during different CHT and DCT strategy, microstructural tests, including scanning electron microscope (SEM) and transmission electron microscope (TEM), were designed to observe the grains and precipitates conditions in the tested samples.

A scanning electron microscope JSM 7200F was used for observations. SiCp–AA2009 composites at as-forged conditions and CHT treated with 20 h ageing time were selected to check the grain evolutions during ageing of the material. The observation samples were 10 mm cubes with the surface polished and electropolished using perchlorate alcohol solution at −12 °C for 20 s [27].

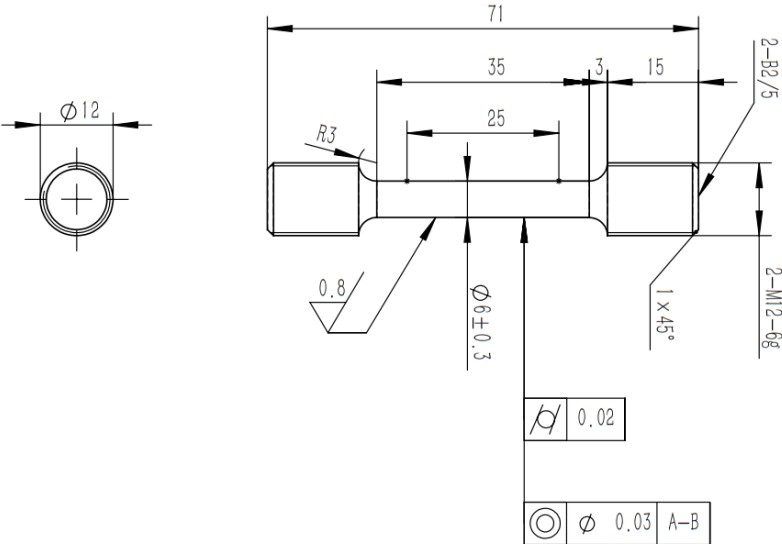

**Figure 2.** Schematic showing the samples for tensile tests (unit: mm).

Transmission electron microscope scanning (TEM) was performed using FEI Tecnai G2 F20, 200KV instrument. Samples after CHT with 4 h ageing (under-aged condition) and 20 h ageing (nearly peak-aged condition), after DCT with 20 h ageing (nearly peak-aged condition) were selected for observations. The samples were cut to discs along the stress direction, mechanically thinned to 0.08 mm and then electropolished at −20 °C [11]. The microscopic precipitates, morphology and characteristics of the samples were obtained. The average precipitates data (sizes, number density) were quantified by taking at least 5 TEM images at different locations of each sample.

## 3. Results and Discussion

### 3.1. CHT and DCT Effect on Aged Mechanical Properties

Figure 3 compares the hardness evolution of SiCp–AA2009 composites and AA2009 aluminum alloy samples along the ageing time. After SHT, a much higher hardness value is observed for the composites (172 HV) than the base material (<131 HV), indicating an apparent strengthening effect from SiCp. The hardness of AA2009 increases continuously for all of the ageing time, indicating that the peak-aged (PA) time should be longer that 60 h for the base material. However, the hardness of SiCp–AA2009 composites increases firstly and then decreases within 60 h ageing, indicating an evolution from under-ageing to PA and over-ageing (OA). The hardness of the composites increases to the maximum value of 188 HV between 20 to 30 h. The hardness of the composites decreases to 163 HV after ageing for 60 h, but it is still higher than the maximum value of the AA2009 based material (147 HV). It is considered that the addition of SiCp can promote the ageing kinetics, leading to a much shorter time for PA, more than 35 h in advance for the SiCp–AA2009 composites investigated in this study. This phenomenon of accelerated ageing is consistent with previously studies reported by Kim et al. [8] and Härtel et al. [10], which has been attributed to the high energy sites provided by the SiCp–aluminum alloy interfaces and the higher dislocation density which is providing more sites for nucleation [28].

Repeated tensile tests have been carried out three times for three testing conditions: the as-received material, and DCT samples with 1 h and 4 h artificial ageing conditions. The standard deviations of yield strength, ultimate tensile strength and elongation values for all three of these testing conditions are within ±5 MPa, ±6 MPa and ±0.8%, demonstrating a good repeatability of the materials and testing conditions. Figure 4 shows the evolutions of UTS, YS, and elongation at fracture (δ) over ageing time obtained by a uniaxial tensile test of SiCp–AA2009 composites with CHT strategy. UTS and YS increase from 487.2 MPa and 285.8 MPa, respectively, significantly in the first couple of hours of ageing,

increased respectively by 37.9 MPa and 69.3 MPa in the first hour. Subsequently, the growth rate decreases significantly and UTS and YS increases gradually to the maximum values of about 555.1 MPa and 472.9 MPa, respectively, at about 40 h, which is considered as the PA states. The results show the same trend with the hardness results before. Then, the sample enters into the OA states, and both UTS and YS begin to decrease slowly. However, for the elongation property of the composite, δ decreases continuously within the 60 h of ageing.

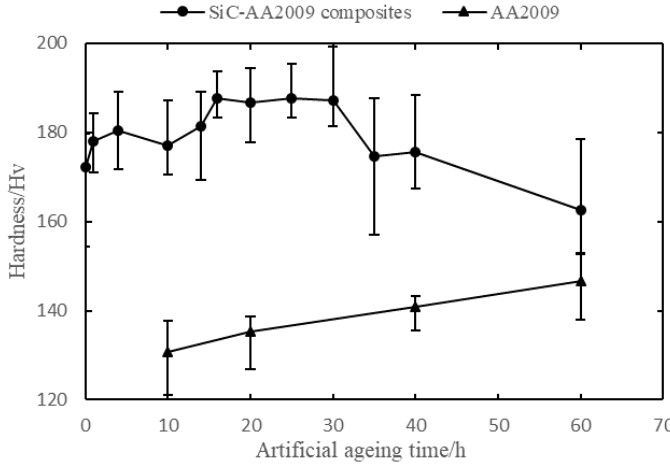

**Figure 3.** Comparison of hardness variations along the ageing time under CHT strategy of SiCp–AA2009 composites and AA2009.

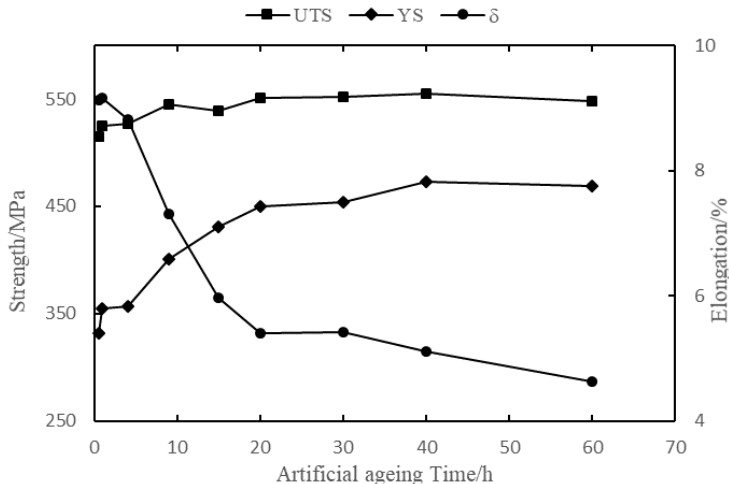

**Figure 4.** Variations of ultimate tensile strength (UTS, left *y*-axis) in engineering stress, yield strength (YS, left *y*-axis) in engineering stress and elongation (δ, right *y*-axis) of SiCp–AA2009 composites aged at 170 °C with different time under CHT strategy.

The tensile test results of aged samples after DCT are shown in Figure 5a; corresponding data from CHT condition are also plotted for direct comparison. The solid lines represent the results of DCT condition, and the dotted lines are the results of the CHT condition. A similar evolution trend of both UTS and YS during ageing are observed under both CHT and DCT conditions, which increase first and then decrease with the ageing time. However, higher strength values are observed with DCT conditions compared to those with CHT conditions under the same ageing time for all the results. The strengths of both the CHT and DCT samples reach the maximum value at the same time of 40 h ageing, while UTS and YS values are, respectively, 11.4 MPa and 27.8 MPa higher for the DCT condition than the CHT condition, as shown in Figure 5b. After that, OA occurs and both UTS and YS values in CHT and DCT samples decrease with further ageing time. Hence, the tensile

test results of DCT condition and CHT condition clearly indicate that DCT can facilitate the improvement of the strength of materials during ageing but has few effects on the strength evolution trend. The related microstructures for this phenomenon will be introduced and discussed in the following section. In addition, the elongation value experiences the same decreasing trend along the ageing time within 60 h for both the DCT and CHT samples, indicating that DCT has a neglect contribution of the elongation properties during ageing at room temperature.

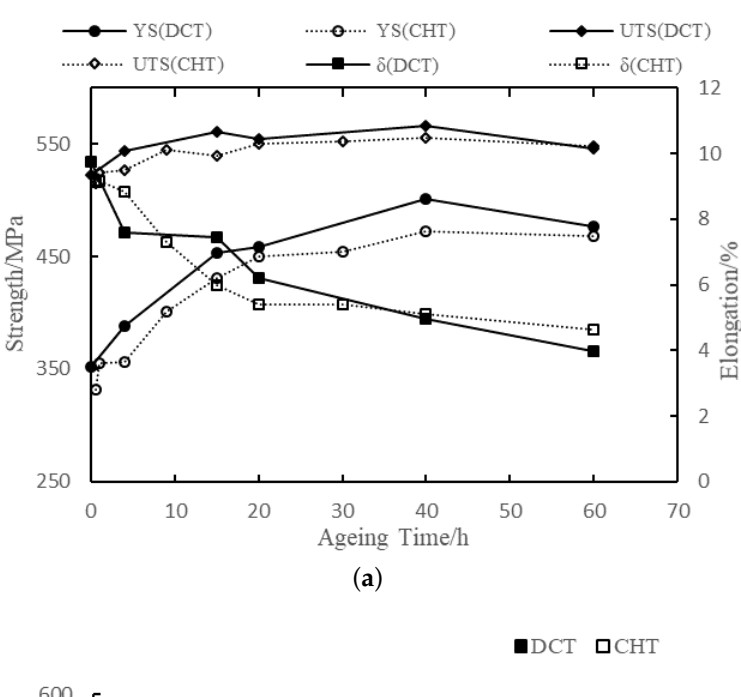

(a)

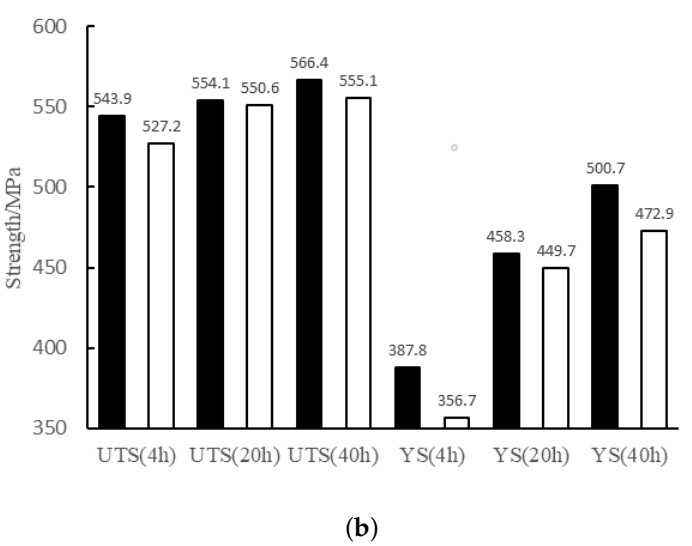

(b)

**Figure 5.** Comparison of (**a**) yield strength (YS, left *y*-axis) in engineering stress, ultimate tensile strength (UTS, left *y*-axis) in engineering stress and elongation (δ, right *y*-axis) variations along the ageing time of SiCp–AA2009 composites under CHT and DCT strategies; (**b**) detailed UTS and YS values comparison under specific ageing times.

### 3.2. CHT and DCT Effect on Microstructures

3.2.1. SiCp and Microstructural Evolution

The microstructures of the composites were observed by SEM at the initial state before heat treatment and in the near PA after 20 h of ageing (Figure 6). The black lumps are SiCp with an average diameter of about 20 μm. SiCp are evenly distributed in the

composites (Figure 6a,b). The shape and size of grains in the composites remain during ageing, as the ageing temperature is not high enough to enable the apparent change of grains [29]. Meanwhile, it can be observed that, after ageing for 20 h, apparent precipitates can be observed in the large scale image of SEM in Figure 6b. Coarse precipitates gathering at grain boundaries resulted in clear grain boundaries compared with initial states. Other needle-shaped precipitates can also be observed inside grains with a uniform distribution, but their sizes are significantly smaller than those at grain boundaries [30]. More details of the precipitate evolution will be discussed in the following sections.

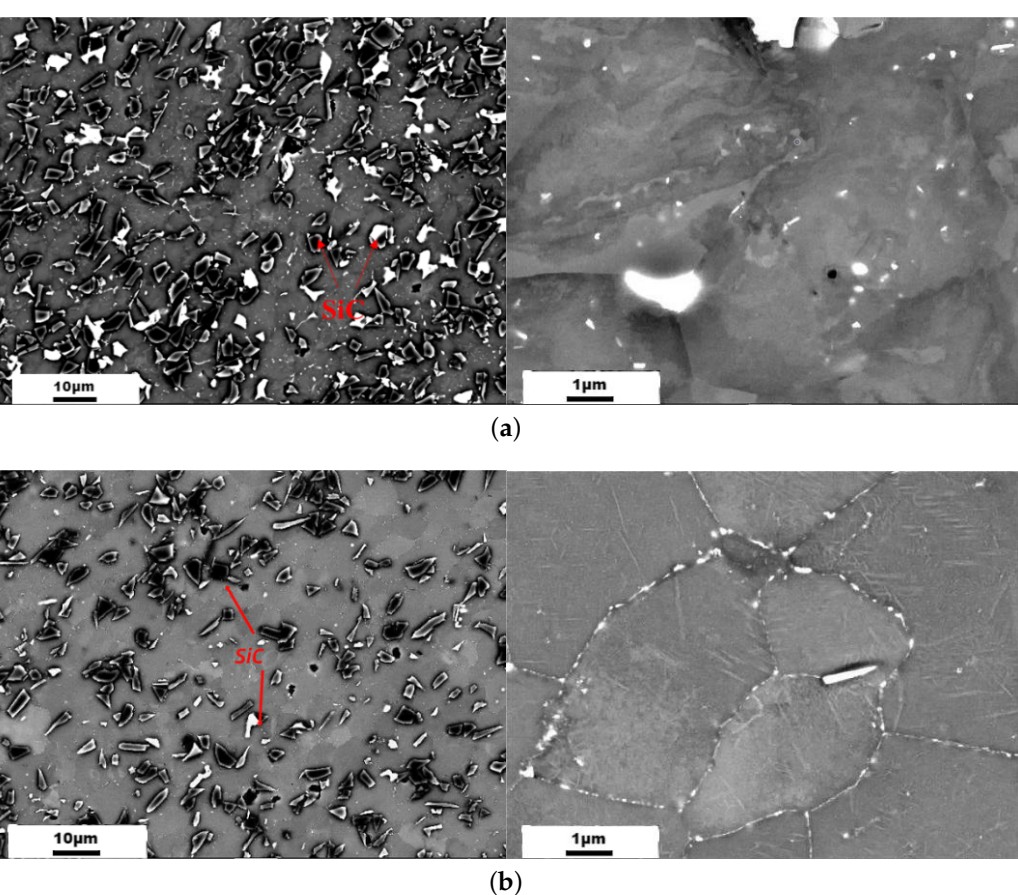

**Figure 6.** SEM images of SiCp–AA2009 composites with different scales under (**a**) initial state and (**b**) near peak-ageing state (20 h ageing) with CHT condition.

### 3.2.2. Intragranular Precipitates

Figure 7 shows TEM images of intragranular precipitates in SiCp–AA2009 composites at the under-ageing condition after CHT with 4 h ageing (CHT-UA), at the CHT-PA condition (CHT with 20 h ageing) and at the DCT-PA condition (DCT with 20 h ageing). The images show that the biggest difference between the samples of CHT-UA condition and CHT-PA condition lies in the density and size of precipitated phase. Sparsely distributed tiny needle-shaped precipitates are observed in the grains at CHT-PA condition in Figure 7a. With further ageing, precipitates further nucleate and growth, leading to the more densely distributed needle-shaped precipitates in a CHT-PA condition in Figure 7b. When comparing the precipitation behavior at DCT-PA and CHT-PA conditions in Figure 7b,c, although the precipitates shape seems similar, an apparent difference in the precipitate size and density can be observed. The needle-shaped precipitates at DCT-PA condition are much thinner and denser compared with the corresponding CHT-PA condition. These denser and thinner precipitates might be the reason for the further improved strength with DCT shown in Figure 5, which will be further discussed in the following sections.

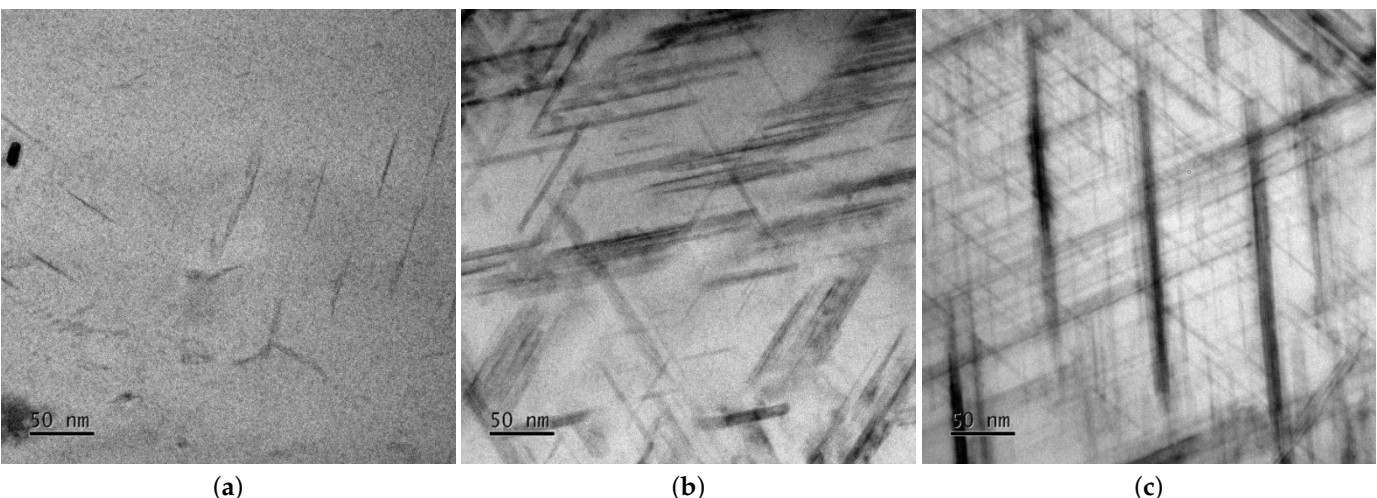

**Figure 7.** TEM images of intragranular precipitates in SiCp–AA2009 composites under (**a**) CHT-UA (CHT with 4 h ageing); (**b**) CHT-PA (CHT with 20 h ageing); and (**c**) DCT-PA (DCT with 20 h ageing).

In order to further analyze the types of precipitates in the above results, HRTEM observations were carried out for specific precipitates, and corresponding FFT patterns were analyzed in <001> Al zone axis, as shown in Figure 8. According to the FFT pattern, the needle-shaped intragranular precipitates are mainly θ′ ($Al_2Cu$) phases [31]. θ′ phases are the main strengthening phase for Al-Cu-Mg alloy, and hence, from 4 h to 20 h, ageing in the CHT condition, the larger and denser θ′ phases in the grains lead to the significant increase of both YS and UTS observed in Figure 4. Figure 8b further indicates that the needle-shaped precipitates in DCT-PA sample are also θ′ phases. Hence, it can be concluded that DCT would not change the precipitation types of the composites but lead to the denser and thinner θ′ phases, contributing to a higher strength effect in the composite material compared with the CHT-PA sample, as shown in Figure 5. Similar results of denser precipitation with DCT condition have also been observed in previous studies for aluminum alloys, and possible reasons have been given that DCT treatment with abrupt temperature changes can increase dislocation density due to the thermal expansion difference between SiCp and aluminum alloys, which could provide more nucleation sites for the precipitates [10,25].

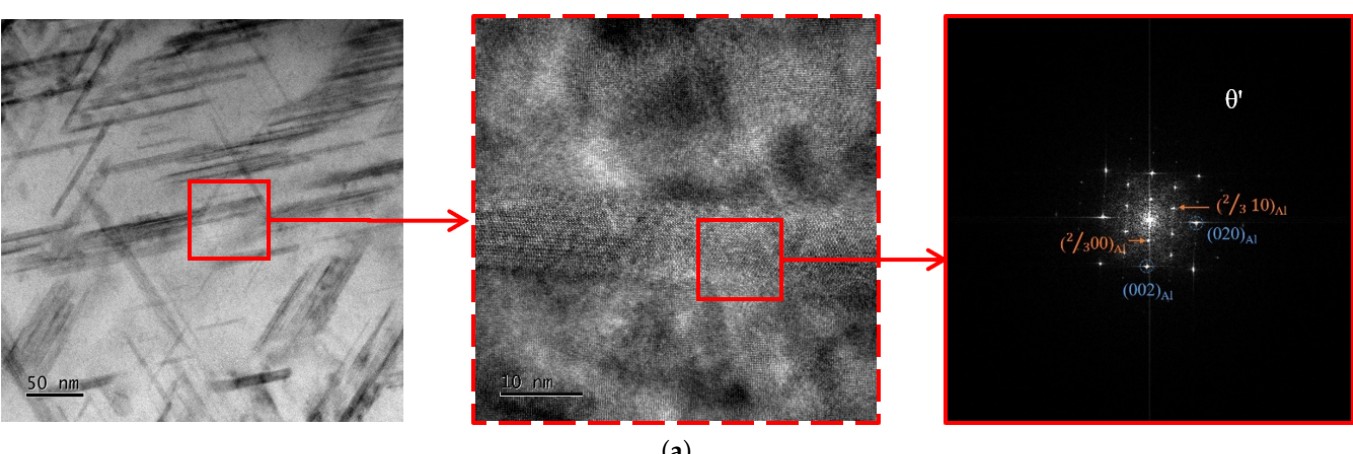

(**a**)

**Figure 8.** *Cont.*

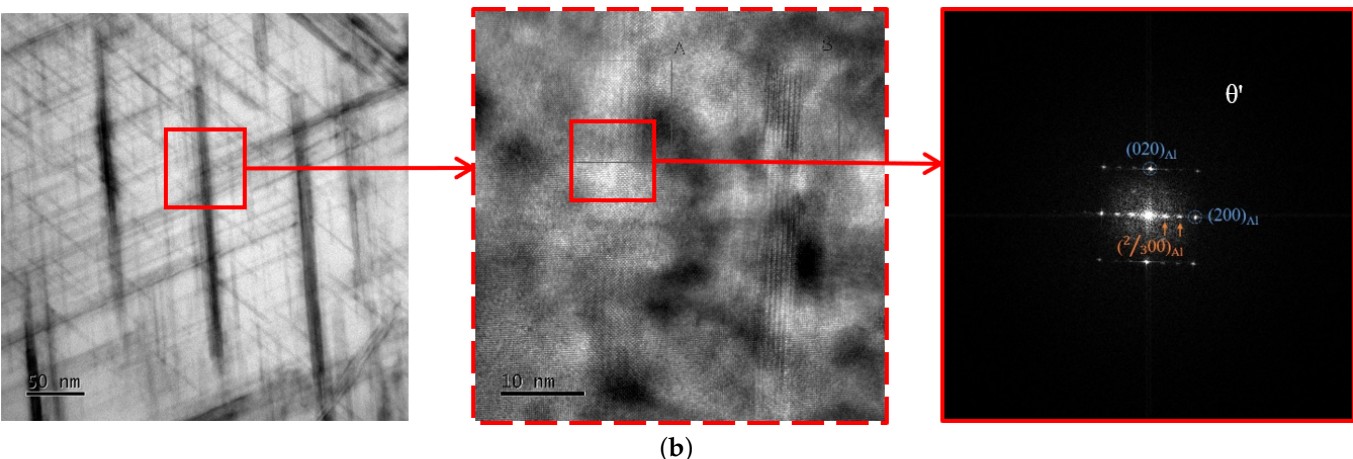

**(b)**

**Figure 8.** TEM and corresponding HRTEM images and fast Fourier transform (FFT) patterns of selected intragranular precipitates in (**a**) CHT-PA sample with identified θ′ precipitates and (**b**) DCT-PA sample with identified θ′ precipitates. The electron beam is parallel to $[001]_{Al}$ axis.

### 3.2.3. Precipitates near Grain Boundaries

TEM images of microstructures in the boundaries between SiCp and matrix alloy are shown in Figure 9. The clean area is SiCp. In contrast to the needle-shaped precipitates in the grain, the precipitates at the grain boundary become larger dot-shaped ones. These dot-shaped precipitates can be observed near the boundaries for all three samples under CHT-UA, CHT-PA and DCT-PA conditions. Unlike the needle-shaped intragranular precipitates with obvious growth trends during ageing, no apparent change of grain boundary precipitates has been observed.

The dot-shaped precipitates are recognized as S phase (Al$_2$CuMg) [32] according to FFT patterns in <001> Al zone axis in Figure 10. S phases are much less likely to occur than θ′ phases due to less Mg atoms than Cu atoms in the alloy. Meanwhile, it is known that S phases contribute less to the strength when compared with θ′ phases [33]. As there is no obvious changes of precipitates near the boundaries under different ageing time and different DCT or CHT processes, the strengthening differences of the material shown in Figures 4 and 5 should be mainly contributed by the intragranular precipitates discussed before.

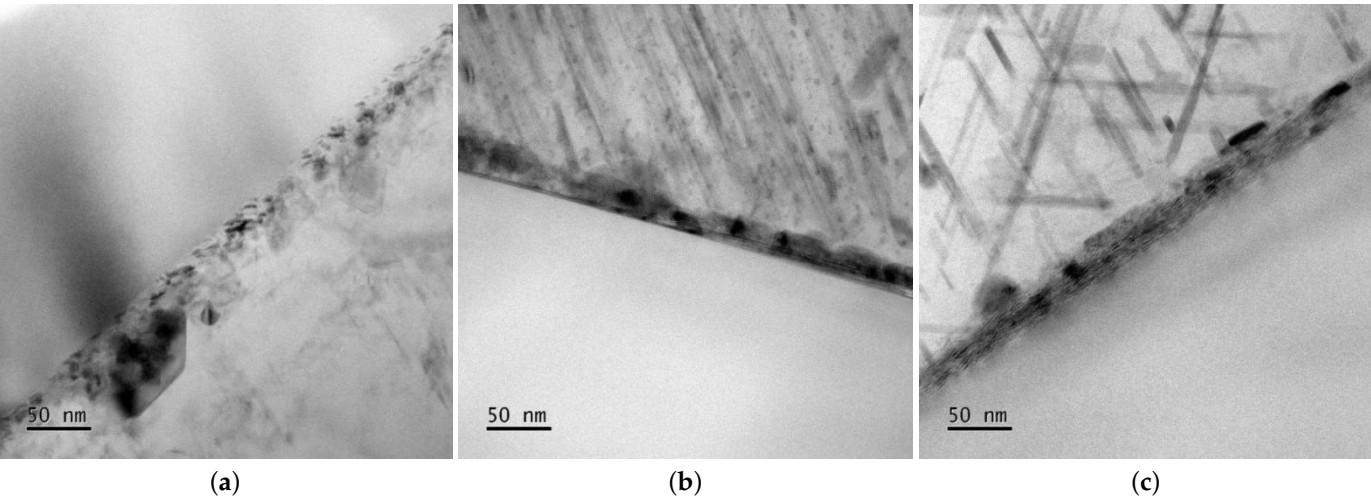

**(a)**            **(b)**            **(c)**

**Figure 9.** TEM images of Precipitates near grain boundaries in SiCp–AA2009 composites under (**a**) CHT-UA (CHT with 4 h ageing); (**b**) CHT-PA (CHT with 20 h ageing) and (**c**) DCT-PA (DCT with 20 h ageing).

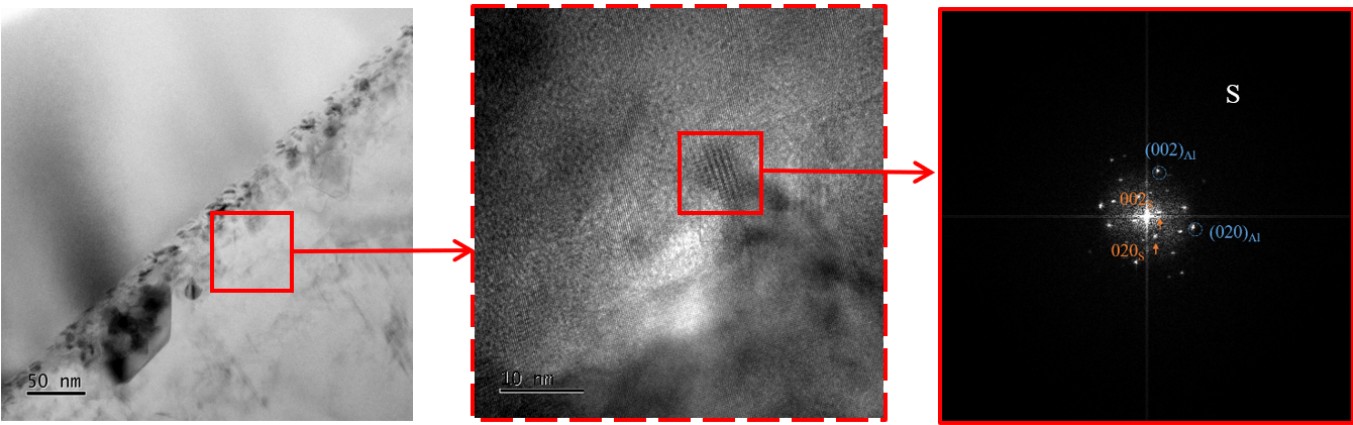

**Figure 10.** TEM and corresponding HRTEM images and fast Fourier transform (FFT) patterns of selected precipitates near grain boundaries in CHT-UA samples with identified S precipitates. The electron beam is parallel to $[001]_{Al}$ axis.

### 3.3. Relationship between Microstructures and Strength

In order to further analyze and validate the mechanisms of enhanced strengths with DCT samples than the CHT samples, a quantified calculation of the yield strength of the composite material is performed in this section. It is widely known that, during ageing of aluminum alloys, the main strengthening contribution comes from the development of second phases, in which the uniformly distributed intragranular precipitates play the key roles in strengthening, while the coarse and heterogeneous precipitates in grain boundaries play a minor role [34]. Hence, only the uniformly distributed needle-shaped θ' precipitates are utilized for the strengthening analysis and calculations in this section. In addition, according to the investigation from Perez et al. [35] , a mean radius methodology that utilizes the average values of microstructures to effectively represent the overall performance of alloys during simple artificial ageing processes is used for calculations in this study.

The contribution of precipitate strengthening in aluminum alloy can be expressed as [36]:

$$\sigma = \frac{M\overline{F}}{b\overline{L}} \tag{1}$$

where $\sigma_p$ is the contribution value of precipitate strengthening strength (MPa). $M$ is the Taylor factor. $b$ is Burger's vector. $\overline{F}$ represents the average resistance to material deformation caused by the existence of precipitates. $\overline{L}$ is the average spacing between precipitates in the material.

As the materials in this study are all before or close to PA state, the shear mechanism is considered as the dominate strengthen kinetics, the average resistance $\overline{F}$ can be expressed by [36]:

$$\overline{F} = k\mu br \tag{2}$$

where $k$ is a constant, $\mu$ is the shear modulus of the materials, and $r$ is the average precipitate radius in the material. The average spacing $\overline{L}$ of the needle-shaped precipitate can be calculated by [37]:

$$\overline{L} = \sqrt{\frac{2}{lN}} \tag{3}$$

where $l$ is the average length of precipitates, and $N$ is the average number density of precipitates.

Meanwhile, the solid solutes experience depletion during ageing, whose contributions to strength can be calculated as [38]:

$$\sigma_{ss} = Kc^{\frac{2}{3}} \tag{4}$$

where $K$ is a strengthening constant, $c$ is the solute concentration in the alloy dependent on the evolution of precipitation, as [37]:

$$c = \frac{c_0 - f_v}{1 - f_v} \tag{5}$$

where $c_0$ is the initial concentration of solution atoms in the alloy after SHT, $f_v$ is the volume fraction of precipitates. For the needle-shaped precipitate, $f_v$ can be calculated by [37]:

$$f_v = \pi r^3 \left( \frac{l}{r} - \frac{2}{3} \right) N \tag{6}$$

For aluminum alloy, its yield strength ($\sigma_{ay}$) is generally composed of intrinsic strength ($\sigma_0$), precipitation strength ($\sigma_p$), and solution strength ($\sigma_{ss}$). The equation is as follows:

$$\sigma_{ay} = \sigma_0 + \sigma_p + \sigma_{ss} \tag{7}$$

Furthermore, considering the discontinuous distributed SiCp in the aluminum alloy based composite, the modified shear lag theory is used to amend the yield strength equations. The final yield strength ($\sigma_y$) of the composite then is expressed as [39]:

$$\sigma_y = \sigma_{ay} \left( v_p \frac{S+2}{2} + v_m \right) \tag{8}$$

where $v_p$ is the volume fraction of SiCp (15% in this study), $v_m$ is the volume fraction of matrix alloys (85% in this study) and S is the shape factor of particles (approximately treated as 1 here). With the further listed equations, the yield strength of the composite materials under different ageing conditions can be quantified.

The related precipitation data, including the length, width, and number density of precipitates, have been measured according to the TEM observations, and the results are listed in Table 3 below.

**Table 3.** Quantified precipitation data from TEM observations.

| Heat Treatment | Ageing Time | Length/nm | Width/nm | Density/$10^{15}\mathrm{m}^{-2}$ |
|---|---|---|---|---|
| CHT | 4 | $44.3 \pm 12.2$ | $3.4 \pm 1.0$ | 0.3 |
| CHT | 20 | $49.0 \pm 22.9$ | $3.6 \pm 1.4$ | 1.2 |
| DCT | 20 | $52.1 \pm 29.3$ | $3.1 \pm 1.9$ | 1.8 |

The material constants for the above equations can be directly obtained from previous literature for artificial ageing of aluminum alloys, whose values used in this study are listed in Table 4. As $\sigma_0$ is generally a constant value and will not change with the change of heat treatment, a value of 10 MPa is given. The material constant $k$ in Equation (2) is calibrated according to the experimental data of CHT sample with 20 h ageing.

**Table 4.** Precipitate strength calculation formula related variables [37].

| Parameter | Value |
|---|---|
| $\sigma_0$ | 10 MPa |
| $M$ | 2 |
| $b$ | 0.286 nm |
| $k$ | 0.11 (calibrated) |
| $\mu$ | 27 GPa |
| $K$ | 840 MPa |

With the microstructural data and material constants listed in Tables 3 and 4, the strength components and final yield strength have been calculated and the results are plot-

ted in Figure 11 below. With the calibrated material constant based on the 20 h ageing CHT sample results, the calculated results for the yield strength of both CHT and DCT samples with 4 and 20 h ageing conditions both agree well with corresponding experimental results. The calculated precipitation strength increases with ageing time due to development of size and numbers of precipitates, which overcomes the weakening effect from solution hardening, leading to the strengthening of the materials. From 4 h ageing with CHT to 20 h ageing, $\sigma_p$ increases from 170.8 MPa to 357.5 MPa, while $\sigma_{ss}$ decreases from 106.7 MPa to 50.1 MPa. For the calculation results of DCT and CHT samples with 20 h ageing, precipitation hardening is enhanced in DCT conditions, showing a 35.3 MPa strengthening effect compared with CHT conditions. The strengthening effects come from much denser and thinner precipitates observed in Section 3.2 before. This strengthening value of 35.3 MPa is much higher than the yield strength difference observed from experiments (about 9 MPa). Meanwhile, more solute depletion is predicted with a further decease of $\sigma_{ss}$ about 25.1 MPa in the DCT sample compared with a CHT sample. Hence, a final yield strength difference between the DCT and CHT treated 20 h ageing samples is calculated as about 10 MPa, which is close to the experimental data of 9 MPa. The theoretical calculation results agree well with corresponding experimental results, which quantitatively support the preliminary conclusion obtained from microstructural observations that the strengthening effect from DCT condition compared with CHT condition mainly comes from the much denser and thinner θ′ precipitates.

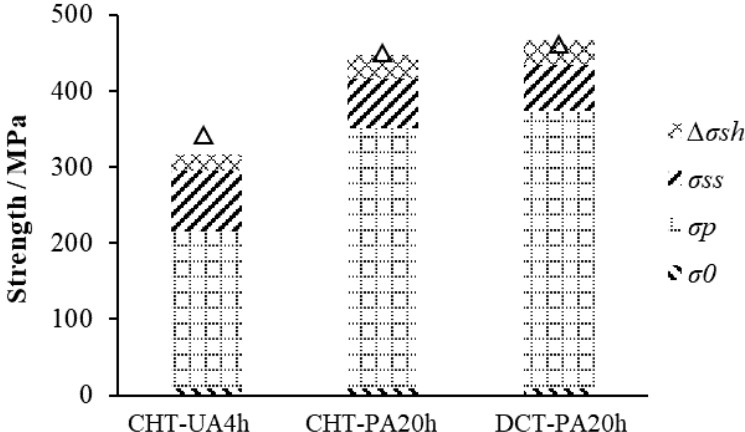

**Figure 11.** Comparison of the yield strength contributions in the samples under different heat treatment from calculations. (The experimental data are plotted as symbols in the figure for comparison).

## 4. Conclusions

The effect of deep cryogenic treatment (DCT) on the artificial ageing properties and corresponding microstructural mechanisms has been investigated experimentally in this study. The following conclusions could be drawn:

1.  The addition of SiCp could significantly promote the ageing kinetics. Peak-ageing of SiCp–AA2009 composites appears more than 35 h earlier than that of AA2009 aluminum alloy (with peak-ageing time of more than 60 h). SiCp contributes a significant strengthening effect in the composite material, and the maximum hardness of the composite (188 HV) is much larger than that of the substrate (147 HV).
2.  DCT will not change the ageing trend of the composite material, but can further improve the yield strength (YS) and ultimate tensile strength (UTS) of SiCp–AA2009 composites compared with those of conventional heat treatment (CHT) strategy. YS and UTS increase by 11.4 MPa and 27.8 MPa, respectively, at the peak-ageing state.
3.  Much thinner and denser homogeneously distributed θ′ (Al$_2$Cu) precipitates are found within the grains of the samples under DCT conditions when compared with corresponding CHT conditions. A theoretical model relating the microstructures and

yield strength of particle reinforced metal matrix composites is utilized to quantify the strengthening effect from the θ′ precipitates, and the quantified results further validate that the strengthening effect in DCT condition mainly comes from the thinner and denser homogeneously distributed θ′ precipitates observed in the composites.

**Author Contributions:** Conceptualization, Y.L.; methodology, software, validation, Z.W. and J.C.; formal analysis, Y.G.; investigation, Z.W. and J.C.; resources, Y.L.; data curation, writing—original draft preparation, Z.W. and J.C.; writing—review and editing, visualization, Z.W.; supervision, B.L.; project administration, R.P.; funding acquisition, B.L. and R.P. All authors have read and agreed to the published version of the manuscript.

**Funding:** The research in this paper was funded by the Natural Science Foundation of Beijing (3214053), Projects for the National Science and Technology Institutes of China (20190101), and the Guangdong Basic and Applied Basic Research Foundation (2019A1515110851).

**Informed Consent Statement:** Not applicable.

**Data Availability Statement:** The data in this paper are obtained from experiments and observations, and the calculated data are consistent with experimental data.

**Acknowledgments:** We acknowledge support from the Institute of Metal Research, Chinese Academy of Sciences (IMR) for supplying materials.

**Conflicts of Interest:** The authors declare no conflict of interest.

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
