# Peer review of "Effect of Deep Cryogenic Treatment on the Artificial Ageing Behavior of SiCp–AA2009 Composite"

_metals, doi:10.3390/met12101767_

Round 1
Reviewer 1 Report
The manuscript metals-1902059 was reviewed.
The manuscript deals with cryogenic quenching of an Al-SiC composite and its effect on microstructure and mechanical properties.
The manuscript is prepared well and suggested for the publication. The authors must address some of the minor comments as following:
1. The scales in TEM images must be enlarged.
2. Equations 2-8 are cited fro mother research studies and do not need to bet mentioned within the manuscript.
Although the topic is not quite novel, but the authors could provide a sufficient characterization in this paper. Especially, the TEM studies in Figs. 7-10. I think some calculations and formula in the manuscript were not very necessary to mention and authors could focus more on tensile stress-strain curves.
Reviewer 2 Report
The authors present experimental results of the effect of deep cryogenic treatment (DCT) of a metal matrix composite (SiCp-AA2009, i.e., aluminium alloy reinforced with SiC particles) on their microstructure and mechanical properties. The experimental results are compared with those obtained by conventional heat treatments. The authors also present some theoretical calculations of the yield strength values estimated from some microstructural observations. They obtain a rather good agreement between the measured and the estimated values. The results are interesting, the experiments results are rigorous and well explained; the discussion is sound; and the manuscript is in general well written.
I have some suggestions to gain in clarity:
- Abstract: I suggest to change “SiCp” by “SiC particles” or “SiC-particles” (line 1).
- Table 2: I suggest to include the names “Ts, TA and TDCT” when giving these values in table 2, to ease comparison with figure 1.
- Table 2: “196” should be “-196”.
- Figures 3 and 4: I suggest to change “Ageing time” by “Artificial ageing time”.
- Figure 4 and 5: What are the errors of these values? Please include errors in the figure.
- Figure 4 and 5a: To be rigorous, please specify by an arrow (or explain in the figure caption) what values refer to both Y axes (left and right).
- Figure 7 also shows some differences on the orientation of the intragranular precipitates between (b) and (c). Could you comment on that?
- Figure 11. Similar comment to above: What are the errors of these calculated values? Are the differences between the values of the yield strength between CHT and DCT higher than the errors (both experimental and calculated values).
- Conclusions, line 296: I suggest to also mention the composition of these precipitates in the conclusions.
- There are some grammar and spelling errors in the article, please revise.
Reviewer 3 Report
Dear Authors,
thank you for sharing your very interesting research and results and for considering Metals as your journal of publishing this research. In general, from my point of view, the results are absolutely worth to publish. The introduction is appropriate which can only be slightly improved by some aspects I will share later. In the second part I have some more remarks. The results and the discussion are also very interesting and pretty well to read. Your conclusion is a brief summary of your results.
As mentioned I have some aspects to consider for a further improvement of your hard work. Please find my comments, question and/or remarks in the order of appearance in the manuscript.
- line 25: Please add after "...and hardness [4] of composites." "Furthermore SiCp may affect the flow behavior in tensile tests by the occurence of Portevin-le Chatelier (PLC) effects [https://doi.org/10.1088/1468-6996/12/6/063001, https://doi.org/10.1016/j.matdes.2014.11.042, https://doi.org/10.3390/met8020088]."
- line 29: please add after "Kim el al. [5]" ", Starink and Gregson [https://doi.org/10.1016/0921-5093(95)10159-4] as well as Härtel et al. [doi:10.1088/1757-899X/63/1/012080] have reported..."
- line 67: "subsequent water-quenching water" --> please remove the second "water"
- line 67: was there an specific time between SHT and AE for CHT? Did you store the specimen at room temperature? We made the experience that even 2 h storing at room temperature already results in precipitations/natural ageing.
- figure 1: can you provide this figure in a better quality? Especially "Quenching in water" and "Quenching in liquid N2" looks very blur and not like optimal resolution of the scheme
- line 76: please add after "...and the hardness..." --> "...hardness measurement..."
- table 2: please add a space for group 1 between "60" and h" and at group 4 between "1" and "h"
- line 85: you were measuring hardness with the Vickers method. Vickers is more appropriate for homogeneous microstructures such as typically in steels. The sharp pyramidal diamond is measuring more small regions and may lead to high deviation in heteregeneous microstructures such as aluminum. Reported standard deviations in figure 3 are confirming this! For aluminum alloys the Brinell method is more recommend where as ball is covering a larger area during the hardness measurement. If possible, please consider another measurement row with Brinell instead of Vickers.
- line 92: the tensile tests were performed at room temperature? Please add this information. Temperatures above and below room temperature have a large impact for this type of MMC (see e.g. https://doi.org/10.3390/met8020088)
- line 98: can you provide more information about the SEM sample preparation of your MMC. This is enormous challenging to avoid scratches from the small SiC particles on the specimen. Your preparation seems pretty nice. That is why I would be interested in more details.
line 98: and an addition: Figure 6 is not an EBSD measurement/image. It is a standard SE image taken in a SEM. EBSD is giving you information about grain orientation and grain sizes and is mostly coloured. Please change the notification of EBSD here already.
- line 104: TEM is clear for me with a FEI with 200 kV acceleration voltage, but you are also showing HRTEM images --> was this performed on the same TEM? Usually HRTEM is conducted with 300 kV acceleration voltage. Please provide more me information and add this in the manuscript if necessary
- line 115ff: Usually hardness values are noted as {value} HV{load in N}. In order that you are using gf instead of N I would recommend to write at least "HV" (with capital letters) instead of "Hv". Thank you
- line 115ff: Furthermore I would suggest to round up the hardness value on full digits. The accurancy that you are suggesting with e.g. "172.3 HV" is not fitting to very optical hardness measurements such as Vickers which strongly depends from the person who is performing those measurements. And also your standard deviations are saying: we are not that accurate like we are suggesting with that value ;) So please only "172 HV" and so on for all values (also in the conclusion)
- line 177: please put a hard space between "60" and "h" that the "h" is not moving lonely to the next line.
- line 126: please change to "...by Kim et al. [5] and [doi:10.1088/1757-899X/63/1/012080", which has been attributed to the high energy sites provided by the SiCp- aluminum alloy interfaces and the higher dislocation density which is providing more sites for nucleation"
- figure 3: if you cannot provide measurements with Brinell method, please note here in the manuscript that the large standard deviations are arising from the Vickers method
- line 128: "elongation" --> which elongation? Elongation without necking? Elongation at failure? --> If elongation at failure, please comment how that elongation is determined --> from the machine? manually with the tested specimen? with or without elastic part?
- line 128: in general I would suggest to print also the results of tensile testing of different conditions to see maybe the serrated flow arising from PLC effects after SHT and the reduction of PLC towards PA condition. This would be another highlight in your manuscript
- figure 4: I miss standard deviations of your values. Can you add them please?
- figure 4: the right y-axis is labeled with "Ductility/%". This is wrong. It is "Elongation/%" (which should also be sharpened in terms of which kind of elongation it is --> see above) --> in figure 5 it is not ductility...
- figure 4: are the strength values in engineering stress or true stress? please add this information in the caption.
- line 138: add "test" --> "The tensile test results..."
- line 148: add "test" --> "the tensile test results..."
- line 154: please add "...properties during ageing at room temperature."
- figure 5a: I miss standard deviations of your values. Can you add them please?
- figure 5a: the right y-axis is labeled with "Elongation/%" --> which elongation?
- figure 5a: are the strength values in engineering stress or true stress? please add this information in the caption.
- figure 5b: please add spaces between "UTS" and "(4)" and for all following beams at the x-axis too please
- line 156 "3.2.1. SiCp and grain microstructural evolution"
- line 157: this is, as mentioned already above, no EBSD. It is standard SE image. For sure, other reviewers mentioned this too
- line 163: not EBSD, just SE
- caption figure 6: not EBSD, just SE
- can you provide any information about the extrusion direction of your material in relation to the microstructural images in SEM and TEM? We made the experience that the precipitation shape and orientation strongly depends from the orientation of the sample
- line 185: "...Al zone axis ,, ,..." (wrong comma)
line 198: please add: ...for the precipitates [20, doi:10.1088/1757-899X/63/1/012080]."
- line 227: "b is Berger's Burgers vector."
line 291: please do a rounding up of the hardness values in the conclusions
Again, overall, your research is worth to publish. My comments and remarks just should further improve the quality. I am sure you can agree in this.
Thank you and best regards.
Round 2
Reviewer 3 Report
Dear authors,
thank you for your efforts. I hope you are seeing the improvement of the manuscript.
All the best and kind regards